# Design and Development of a Reduced Form-Factor High Accuracy Three-Axis Teslameter

**Johann Cassar [1],\*** , **Andrew Sammut [1]** , **Nicholas Sammut [2]** , **Marco Calvi [3]** ,
**Sasa Dimitrijevic [4]** and **Radivoje S. Popovic [4]**

[1]   Faculty of Engineering, University of Malta, MSD 2080 Msida, Malta; andrew.sammut@um.edu.mt
[2]   Faculty of ICT, University of Malta, MSD 2080 Msida, Malta; nicholas.sammut@um.edu.mt
[3]   Photon Science Division of the Paul Scherrer Institute, 5232 Villigen PSI, Switzerland; marco.calvi@psi.ch
[4]   Sentronis ad, 18115 Nis, Serbia; sasa_dimitrijevic@sentronis.rs (S.D.); rpopovic@senis.ch (R.S.P.)
\*   Correspondence: johann.cassar@um.edu.mt; Tel.: +356-2340-3339

**Abstract:** A novel three-axis teslameter and other similar machines have been designed and developed for SwissFEL at the Paul Scherrer Institute (PSI). The developed instrument will be used for high fidelity characterisation and optimisation of the undulators for the ATHOS soft X-ray beamline. The teslameter incorporates analogue signal conditioning for the three-axes interface to a SENIS Hall probe, an interface to a Heidenhain linear absolute encoder and an on-board high-resolution 24-bit analogue-to-digital conversion. This is in contrast to the old instrumentation setup used, which only comprises the analogue circuitry with digitization being done externally to the instrument. The new instrument fits in a volumetric space of 150 mm × 50 mm × 45 mm, being very compact in size and also compatible with the in-vacuum undulators. This paper describes the design and the development of the different components of the teslameter. Performance results are presented that demonstrate offset fluctuation and drift (0.1–10 Hz) with a standard deviation of 0.78 µT and a broadband noise (10–500 Hz) of 2.05 µT with an acquisition frequency of 2 kHz.

**Keywords:** analogue-to-digital conversion; ATHOS soft X-ray beamline; broadband noise; Hall probe; offset fluctuation and drift; three-axis teslameter; undulator

## 1. Introduction

Various parameters are typically taken into consideration in the measurement of magnetic fields and primarily include the field range, the bandwidth, the reproducibility and the accuracy [1,2].

Whilst static or quasi DC homogenous magnetic fields are best measured using NMR (Nuclear Magnetic Resonance) instruments with the best absolute accuracies, these have a very limited bandwidth [1]. The choice of silicon-based Hall magnetometers is very attractive when a balance between the precision and the bandwidth response of the instrument is required as is the case in this application. Also, three-axes Hall probes fulfill the necessity in measuring all three components of a magnetic field simultaneously along a straight line as required in undulators.

The measurement with Hall probes is particularly suitable for a broad range of non-homogenous magnetic fields up to 10 T and more. However, the absolute accuracy of the measurement is affected by parasitic effects such as nonlinearity, temperature dependence, planar Hall effect and stability of the offset [1–3]. Hence apart from calibration, these inherent problems in Hall probes require compensation through special biasing and interfacing techniques.

The small and compact volume of the Hall probe allows for high spatial resolution of the magnetic measurement. This makes them very suitable and the preferred choice in mapping the magnetic

field in insertion devices (such as undulators) especially when these have a very narrow gap and the bandwidth requirements are not particularly high.

Magnetic field mapping of undulators sets the context of the main application of the development of the three-axis teslameter presented in this paper. In [4], a Hall-probe bench for insertion device characterization at the Brazilian Synchrotron Light Laboratory (LNLS) is presented whereby three single-axis Hall probes are orthogonally assembled and used for the magnetic field component measurements. The use of Hall probes as measuring devices are preferred over rotating coil techniques in this scenario. For such a setup the dimensions of the sensor must be significantly smaller than a single undulator period for proper mapping to be done. This makes Hall probes very advantageous to use over short moving coil techniques. Through these measurements, the existence of random local angular kicks along the undulator axis can also be investigated. Post-processing of the data results in the measurement of the phase error along the undulator axis. On-the-fly feedback scanning is also essential to reduce the sensor vibration during traversal motion. A high level of accuracy in the sensor position on the longitudinal axis is also critical, as pointed out in [2].

Other techniques are used in order to measure the integral field errors along the undulator. These give information on the total change in the angle and position of the beam trajectory at the exit of the undulator. In [5], a method is presented where the field integrals are measured using a multistrand wire stretched inside the undulator. As the wire is moved with constant velocity of translation, the first integral is found. The cross motion of the wire at the undulator ends measures the second field integral.

Therefore, knowledge of these errors is very important for a complete characterization of the magnetic field in an undulator. After being measured, the errors are corrected by magnets shimming as explained in [6].

Following this background on the problem definition, the rest of the paper tackles the development of a novel three-axis teslameter interfaced with a SENIS type-S Hall probe [7]. This instrument will be used with a new magnetic measurement bench developed at the Paul Scherrer Institute (PSI) for the highly precise magnetic field measurements of the ATHOS soft X-ray line [8].

The main motivation of the development of this instrumentation is to improve the current instrumentation used [9] to characterize the ATHOS line of undulators. The main aim is to integrate the full analogue and digital electronics on a single board with a tenfold size reduction and increase the performance and features compared to the old instrumentation. Another goal is to provide an instrument that can easily be used for similar applications in other machines hence it is also important to provide a more cost-effective and maintenance-free setup.

## 2. Hall Probe Theory

As explained in [10], the resolution of a magnetic sensor depends on its intrinsic noise, offset instability and the magnetic sensitivity. Silicon Hall sensors are typically realized as an n-well plate with four contacts being easily modelled as a Wheatstone bridge [11,12]. Since a current is made to flow through two opposing terminals, a magnetic field perpendicular to the plate makes electrons drift towards one side thereby generating a potential difference known as the Hall voltage.

Due to doping inhomogeneity and variations in the depth of the n-well, the resistance in the various branches of the Wheatstone bridge usually do not match, thus the sensor will exhibit a certain amount of offset when biased as shown in Figure 1. The effect of the offsets can be significantly reduced by employing the spinning current technique [3,11,12]. Swapping the functions of the readout and bias electrodes and thus changing the direction of the bias current through the sensor, swaps the relative polarities of the sensor's offset and Hall voltage. Therefore, doing this periodically, results in the offset being modulated to the spinning frequency while the Hall voltage is recovered by averaging the voltage on the other two contacts.

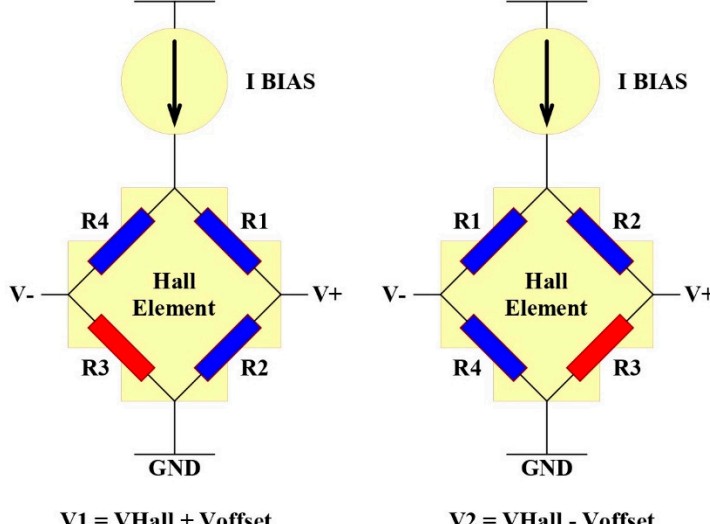

**Figure 1.** Wheatstone bridge model of a Hall sensor biased with a constant current source and exhibiting an offset voltage.

When exposing the Hall probe to a nonorthogonal magnetic field, the output Hall voltage appears to be the sum of the normal and planar Hall voltages. The planar Hall voltage is dependent on the magnetic field and appears also as an offset voltage where it is proportional to the square of the planar field component. The spinning current method also suppresses the planar Hall Effect [3].

A challenge in the application of the spinning current technique as explained in [10] is the resultant switching spikes that occur upon current reversal. Unfortunately, these spikes result in additional noise, offset drift and $1/f$ noise that must addressed properly in the electronic design since they impact the overall performance of the magnetic measurements. This is the main limitation that makes it difficult to approach the physical limit of the magnetic resolution of the Hall device. High-order low-pass filtering circuitry must be used in order to attenuate sufficiently high order harmonics generated by these spikes. This is tackled in the actual design of the instrument as explained in Section 3.

## 3. Architecture and Components of the Teslameter

This section provides a detailed overview of the complete architecture of the newly developed teslameter with all details pertaining to the circuitry involved and the interfacing operation capabilities of the instrument. The advancement and the state of the art of this instrument is the incorporation of all the analogue and digital circuitry as explained in this section all on a single printed circuit board (PCB) with very tight dimensions. In addition, this instrument also consists of a digital interface to a Heidenhain linear absolute encoder thus making it possible in providing synchronized position and magnetic field readings. This instrument provides a tailored solution for undulator mapping applications with better performance in both synchronization timing and noise performance, as well as a more complex, flexible and hence precise and accurate calibration routine to be applied.

### 3.1. Architecture

A block diagram of the instrument architecture is presented in Figure 2. The indicated analogue circuitry consists of identical spinning current and voltage readout circuitry for the interfacing of a three-axis SENIS H3A Hall probe [7]. The analogue differential voltage of each axis is equally amplified using a very low noise instrumentation amplifier and low pass filtered using a third order Butterworth antialiasing filter to the required 500-Hz bandwidth. The signal path is kept fully differential down to the AD converter (ADC). The 24-bit, 4-Ch, simultaneous sampling Delta-Sigma analogue-to-digital converter is the core of the instrument that provides the necessary digitization of the three magnetic field axes and the Hall probe temperature PT100 analogue reading.

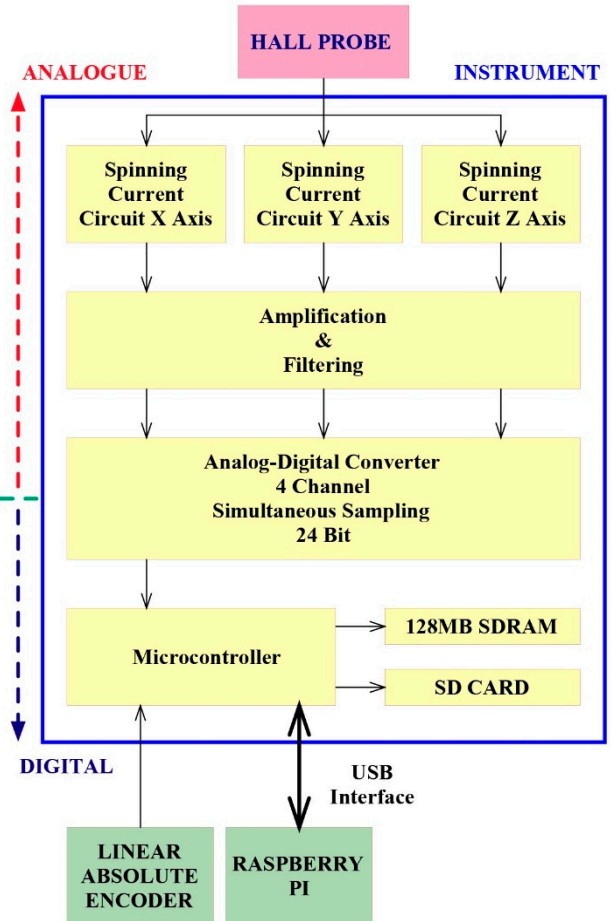

**Figure 2.** Architecture block diagram of the whole instrument showing the different functions in the analogue and digital sides.

The digital part is organized around the C2000 series Delfino TMS320F28379D microcontroller [13]. This powerful 32-bit floating point and dual core industrial microcontroller is designed for advanced closed-loop control applications and provides 200 MHz of signal processing performance in each core. The microcontroller handles all the communication interfaces with all the peripherals explained further in the next section.

The instrument circuitry board is an eight-layer PCB with physical dimensions of 144 mm in length by 44-mm wide. The eight-layer choice was deemed to be the best option considering the circuitry complexity involved and the physical size restriction that the instrument must fit in. A top view photo of the PCB is presented in Figure 3. The PCB layer stack-up choice incorporates two middle signal layers which are sandwiched between two ground planes for superior electromagnetic compatibility (EMC) performance in comparison to a six-layer stack up choice mainly due to the additional ground plane. The two ground planes enable the incorporation of ground-to-ground vias between the two ground planes near the signals vias in order to provide an adjacent return path for the current.

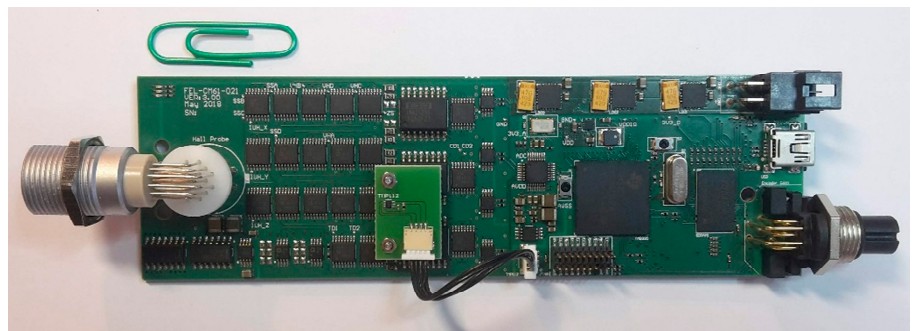

**Figure 3.** Top view of the eight-layer PCB, the left side comprises the analogue circuitry with the Hall probe connector at the extreme left, the ADC and the microcontroller somewhat in the middle of the board and the power supply connector, USB and encoder connector on the right hand side of the board.

These two signal layers are used mainly for the routing of the very sensitive and noise prone analogue tracks and high speed signal tracks in the digital section of the board. Adjacent to the two ground layers, a power plane layer is included on each side to accommodate the positive and negative power supply rails. The general routing of the board is done through the two outermost top and bottom signal layers. As copper planes are present beneath the outermost signal layers, ground return paths are always present, minimizing crosstalk and distortion between adjacent signal tracks.

In such a mixed signal PCB, proper design considerations are taken for the proper layout of the ground plane. Failure to do so implies a degradation in the noise performance of the instrument as the digital switching noise couples with the sensitive analogue signals. A void slit in the ground plane isolates the analogue precision circuitry and only a single star point beneath the ADC provides the necessary high impedance connection between the analogue and the digital ground planes. This minimizes any unwanted ground return paths from the digital side to enter in the analogue side [14–16].

### 3.2. Instrument Components

### 3.2.1. Spinning Current Circuitry

In the spinning current modulation process, the current through the Hall plate is injected alternatively along the north–south or the west–east arms. The current direction and voltage readout from the four-terminal Hall plate is controlled using the ADG1612 CMOS analogue switches. These switches have a nominal on-state resistance of 1 Ω [17]. The flatness of the on-state resistance being defined as the difference between the maximum and minimum values of on-resistance as measured over the specified analogue signal range is 0.2 Ω. As this change in resistance over the range introduces distortion, these switches were chosen for their very flat profile thus ensuring excellent linearity and low distortion. The switching of the CMOS analogue switches are controlled using synchronized pulse width modulation (PWM) control signals from the microcontroller.

Jitter-free operation is achieved through the use of hardware implemented PWM modules within the microcontroller. Once configured these free running PWM modules require little overhead and do not interfere with the operation of the microcontroller in data acquisition. Synchronization of the current direction switching and the voltage readout switching is ensured by setting one of the modules with a master time base and the downstream modules are elected to run in synchronization with the master.

### 3.2.2. Spinning Output Amplification

An attentive noise analysis study was conducted for the best possible choice of the instrumentation amplifiers which ultimately limits the noise floor of the instrument. The two most important factors that outline the figure of merit for the amplifier's noise voltage and current response are flicker noise, which is mostly dominant at low frequencies, and shot noise being dominant at frequencies beyond

the corner frequency. In the analysis presented, the measurements are referred to the amplifier inputs in order to remove the need to account for the amplifier's gain.

This Hall plate has $R_{out}$ = 300 Ω with *4kT.$R_{out}$* thermal noise which amounts to 2.222 nV/$\sqrt{\text{Hz}}$. The noise level generated by the amplifier current noise is 90 pV/$\sqrt{\text{Hz}}$ due to the source resistance. Therefore the summation of the noise from the source resistance is 2.223 nV/$\sqrt{\text{Hz}}$. From this it is clear that due to the low source resistance, the voltage noise dominates over the current noise. As the additional noise voltage brought by the op amp is 8 nV/$\sqrt{\text{Hz}}$ at a bandwidth of 1 kHz [18] this results in a total noise spectral density of 8.303 nV/$\sqrt{\text{Hz}}$.

The gain of the amplification stage is set to 4.4 and the bandwidth of the low pass filter is set to 500 Hz with a brick wall correction factor of 1.57. Therefore, the noise voltage root mean square (RMS) value at the output of the amplifier where $e_t$ is the total noise spectral density of 8.303 nV/$\sqrt{\text{Hz}}$ is given by Equation (1):

$$V_t = G \times e_t \times \sqrt{BW} = 1.023 \times \mu \times V_{rms} \tag{1}$$

Therefore, by combining the different component noise sources both from the sensor and the op amp, the noise floor at the output of the amplification stage is approximated to be $1.023 \times \mu \times V_{rms}$.

### 3.2.3. Current Source Hall Probe Biasing

The sensitivity response of the Hall element depends on the current magnitude passed through it. Some factors must be considered in choosing this current magnitude, as the lower the current is, the more gain one has to apply in order to achieve the desired full dynamic range. Higher gains generally imply higher noise figures, so a tradeoff was found experimentally in determining the best excitation DC current. However higher currents passed through the Hall element automatically result in higher sensitivity dependence on the probe temperature. A 2.5 mA Howland Current Source circuit [19] is used to drive each axis of the probe. This current source relies on a very high precision 2.5 V reference [20] being buffered to the Howland op amp circuit to avoid any loading and drifts of the reference voltage. One of the main factors that are considered in the design of this current source is the matching of the four resistors attaining the negative feedback of the op amp which is fundamental otherwise a dependence of the output current on the load magnitude occurs. For this reason, a tightly matched resistor array with a very low temperature coefficient was used for each current source.

### 3.2.4. Interfacing of the Hall Probe PT100

Readout of a voltage proportional to the Hall probe temperature is implemented by passing a constant current through the on chip PT100 whose resistance varies linearly with temperature. This is necessary in applying proper calibration to the magnetic field readings from the Hall probe. Minimization of the sensor self-heating is ensured by passing a considerable low current of 0.25 mA. Interfacing to the platinum PT100 is done using a four-wire configuration rather than two-wire. This allows elimination of the effect of lead resistance as only the very low input bias current of the differential op amp passes through the two voltage readout terminals [21].

For proper temperature readouts, the nonzero offset voltage of the amplifier can be a problem as this drifts with time and temperature. In order to overcome this problem, the offset voltage is measured by reversing the current through the PT100 at a fixed frequency, in this case being 7.8 kHz.

When the current is reversed, the voltage due to the sensor reverses sign while thermal EMFs do not [22]. By averaging the forward and reverse current voltage measurements, the error in the voltage measurement due to thermal EMFs is thus eliminated.

### 3.2.5. Antialiasing Filter and Analogue to Digital Converter

The signal chain is kept fully differential from the Hall plate output to the ADC input. This provides increased immunity to external noise and better signal-to-noise ratio (SNR) performance. Also, a reduction in the even order harmonics is registered and a doubling in the dynamic range for

the same voltage swing when compared to a single ended system is achieved. Figure 4 shows the differential signal levels at the Hall plate output and after amplification. For easier implementation the common mode voltage of the signal chain is kept tied to ground so as to avoid the introduction of additional offset voltages.

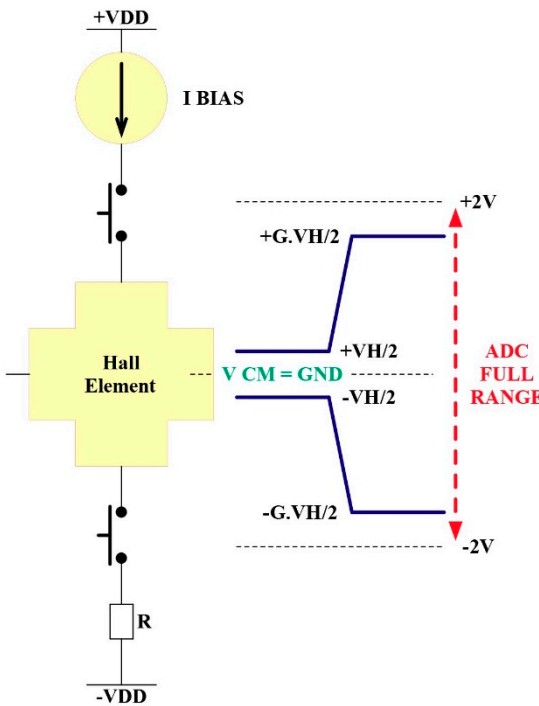

**Figure 4.** The differential signal levels at the Hall element output and after amplification.

A third-order low-pass Butterworth antialiasing filter with a designed bandwidth of 500 Hz, a quality factor of 0.707 and unity gain was designed and implemented for the necessary reduction of the out-of-band noise and minimization of distortion by matching the filter's output to the input circuitry of the ADC.

The multiple feedback circuit topology as shown in Figure 5 makes use of the two complex pole pairs in the feedback chain to set the desired cut-off frequency at 500 Hz whereas the third real pole at the output is set at a higher value in order to sum up an attenuation of −60 dB in the stop band [23].

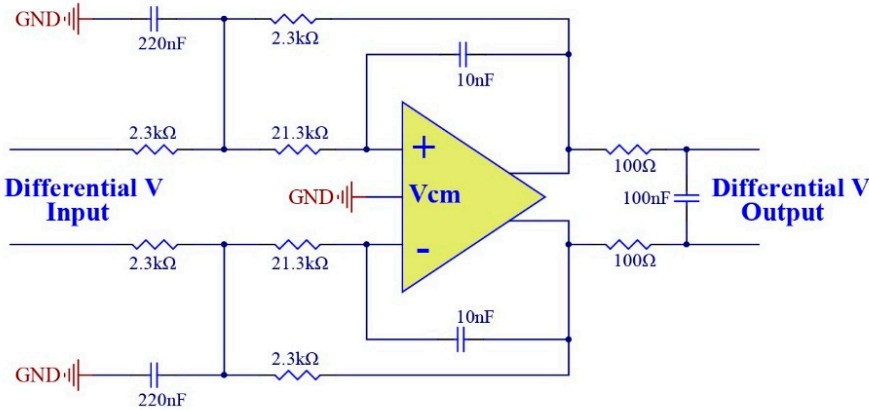

**Figure 5.** Third-order low-pass Butterworth differential antialiasing filter.

As the Butterworth filter topology offers the "maximally flat" response in the passband with the steepest roll-off, it was the preferred choice over other filter topologies.

Physical implementation of the Butterworth filters on the PCB involved optimal position placement of the passive components around the THS4131 differential op amp [24] in order to make the circuit as compact as possible and minimize the length of all trace runs for perfect symmetrical paths and minimal stray inductance pickup.

The antialiasing filter fully differential output is fed in the ADS131A04 24-bit delta-sigma analogue-to-digital converter [25]. The data rate flexibility, wide dynamic range and interface options makes this device well-suited for industrial and instrumentation applications where high precision digitization is required. A bipolar supply of $\pm$2.5 V is used to power up the analogue side and a separate power supply of +3.3 V derived from a separate low-dropout regulator (LDO) powers the digital side of the ADC. General EMC design guidelines were followed to keep the system noise as low as possible.

The external reference voltage (+4.096 V), which sets the input signal range, is derived from a separate reference source with heavy decoupling. This reference voltage is also internally buffered in the ADC for minimal loading effects.

The output data rate adjustment offers a tradeoff between the noise performance and the data acquisition frequency. When averaging is increased by reducing the data rate, noise drops considerably. As the system bandwidth is 500 Hz, the minimal data rate to be used according to Nyquist Sampling Theorem is 1 kHz.

The effective resolution defined for this ADC at the optimal noise performance data rate of 1 kHz is 22.19 bits. This is determined experimentally by shorting the analogue inputs together and taking an average of multiple readings across all channels. One second of consecutive readings are used to calculate the RMS noise [25]. The internal gain amplifier of the ADC is set to unity gain as this degrades the effective resolution. Equation (2) shows the relationship between the effective resolution and the RMS noise.

$$Effective\ Resolution = log_2\left(\frac{2 \times V_{REF}}{Gain \times V_{RMS}}\right) \tag{2}$$

Therefore 1.66 $\mu V_{rms}$ results in 22.19 bits effective resolution. The analogue inputs of the ADC are directly connected to a switched-capacitor sampling network in an unbuffered mode. The ADC does not include any input buffers as these would induce input noise thus lowering the resolution. The capacitors of the switched capacitor input delta-sigma ADC are continuously being charged and discharged at the modulation sample frequency. Because the internal capacitors must be very small when compared to the external circuitry, the average input impedance of the ADC appears to be resistive. At a modulation frequency of 4.096 MHz and internal capacitor values of 3.5 pF, Equation (3) gives an input impedance of approximately 130 k$\Omega$.

$$Z_{IN} = \frac{2}{f_{MOD} \times C_S} \tag{3}$$

The internal architecture of the delta-sigma ADC consists of a modulator at the input which samples the input signal at the rate of $f_{MOD}$. The modulator then converts the analogue input voltage into a pulse-code modulated (PCM) data stream. The sinc$^3$ digital filter takes this bitstream and provides attenuation to the now shaped higher frequency noise [25].

The magnitude frequency response of the sinc$^3$ filter has notches (or zeroes) that occur at the output data rate and its multiples. At these frequencies the filter has infinite attenuation. The sinc$^3$ filter magnitude frequency domain transfer function is given by Equation (4). As *N* is the decimation ratio which varies according to the set output data rate, the theoretical bandwidth of the filter depends on this frequency. Table 1 shows the filter's bandwidth for the set output data rates.

$$|H(f)| = \left|\frac{sin\left[\frac{N \times \pi \times f}{f_{MOD}}\right]}{sin\left[\frac{\pi \times f}{f_{MOD}}\right]}\right|^3 \tag{4}$$

**Table 1.** The sinc$^3$ filter bandwidth for set values of the output data rate.

| Output Data Rate/kHz | Sinc$^3$ Filter Bandwidth/Hz |
|:---:|:---:|
| 1 | 262 |
| 2 | 524 |
| 4 | 1048 |
| 8 | 2096 |

It can be seen that the desired 500 Hz bandwidth of the Hall probe is compromised at an acquisition frequency of 1 kHz. Predominantly due to the inclusion of the digital filter in the delta-sigma ADC chain, a calculable time delay between the analogue input and the digital output is present. This time delay is composed of the delay caused by the digital logic for the ADC to determine whether its conversions are synchronized. For this reason, the digital filter output is placed in a buffer for an entire conversion cycle before it is output. The second component of the delay is due to the group delay by the linear phase response of the sinc$^3$ filter as explained in [26].

The group delay of the sinc$^3$ filter is defined by Equation (5) where $D$ is the decimation rate and $f_M$ is the modulation frequency.

$$\tau_D = \left(\frac{D-1}{2}\right) \times \frac{3}{f_M} \tag{5}$$

Due to its linear phase response, the sinc$^3$ filter does not introduce additional distortion as no matter what the input frequency is, the output is always delayed by the same number of samples. Table 2 provides a breakdown of the group delay times for each output data rate.

**Table 2.** Breakdown of the time delays from the sinc$^3$ filter and digital logic of the ADC for set values of the output data rate.

| Output Data Rate | 1 kHz | 2 kHz | 4 kHz | 8 kHz |
|:---:|:---:|:---:|:---:|:---:|
| OSR | 4096 | 2048 | 1024 | 512 |
| Number of Samples Delay from Sinc$^3$ filter/samples | 1.4996 | 1.4993 | 1.4985 | 1.4971 |
| Number of Samples Delay from Digital Logic/samples | 1 | 1 | 1 | 1 |
| Total Number of Samples Delay/samples | 2.4996 | 2.4993 | 2.4985 | 2.4971 |
| Total Group Delay/ms | 2.4996 | 1.2496 | 0.6246 | 0.3121 |

Device communication is attained using a 20 MHz serial peripheral interface (SPI). The ADC is operated in continuous conversion synchronous master mode operation whereby the ADC signals the microcontroller of a complete data conversion by a negative edge trigger on the data ready (DRDY) line. The ADC then clocks out the last conversion data upon reception of the clock.

As data is transferred in 32-bit packet sizes and the analogue converted data is 24 bits long, the last 8 bits transferred of each packet are Hamming code validation bits. Calculation of the Hamming code on the received data is carried out by a software algorithm for each received packet and the data packet is scrapped in the scenario that the code received and calculated are not identical. Therefore, the Hamming code calculations on the digital interface to the ADC enhance the integrity of the communication channel.

### 3.2.6. Microcontroller and SDRAMs

The transfer of the acquisition data from the ADC is stored on board a 128 MB synchronous dynamic random access memory (SDRAM) during measurement time. The external memory interface module (EMIF) of the microcontroller supports a 32-bit data interface to four bank SDRAM devices. Communication is attained using the internal direct memory access module (DMA) of the microcontroller, which provides a hardware method of transferring data between peripherals and or/memory without intervention from the central processing unit (CPU), thereby freeing up the

bandwidth of the CPU for other system functions. In this way, the ADC interrupt signal timing is not disrupted.

As 64 MB off-the-shelf SDRAM chips are available, two IS42S16320D-6BLI memory chips [27] are routed to the microcontroller whereby high-speed routing design techniques are applied for optimal performance.

The routing architecture of the two 54-ball TF-BGA SDRAM chips to the microcontroller assumes a symmetrical tree layout coupled with minimal clock skews between the command/address/control buses and the data bus [28].

An overall timing budget as explained in [29] was performed in order to determine the data-valid window considering a 100 MHz clock. The timing budget starts with the full cycle time allowed, in this case a 10 ns clock. As a general rule of thumb the total skew between the data lines should fall under 5% of the clock period, which gives an interval of 0.5 ns.

As the propagation delay of a microstrip line is given by Equation (6) in [30] and for a PCB with a typical dielectric constant $\varepsilon_r$ of 4.4, the microstrip's delay constant results in 5.44 ps/mm.

$$t_{pd}(ps/mm) = \frac{85}{25.43} \times \sqrt{0.45\varepsilon_r + 0.67} \tag{6}$$

Therefore, the board skew for an interval of 0.5 ns is 91.69 mm. Matching the data trace lengths to a maximum length difference of 91.69 mm is not a problem in this case, as all trace lengths from the microcontroller to the SDRAM chips are all less than 60 mm in length. The transmitter and receiver skews are obtained from the device's data sheets and included in the timing budget.

As the high speed routing of the memory lines is mainly done from the two innermost signal layers which are sandwiched between two ground planes, micro strip transmission line theory is applied for the correct calculations of characteristic impedance and termination impedance matching [30].

For a signal trace of width $W$ and thickness $T$, separated by distance $H$ from a ground (or power) plane by a PCB dielectric with dielectric constant $\varepsilon_r$, the characteristic impedance is defined by Equation (7) with all measurement dimensions in mils.

$$Z_O(\Omega) = \frac{87}{\sqrt{\varepsilon_r + 1.41}} \times \ln\left[\frac{5.98 \times H}{(0.8 \times W + T)}\right] \tag{7}$$

Given that the trace width $W$ is 0.1 mm, trace thickness $T$ is 35 μm, dielectric thickness $H$ of 135 μm and a relative dielectric constant $\varepsilon_r$ of 4.4, the characteristic impedance of the traces is found to be 70.33 Ω.

Termination of the driver's output impedance to the transmission line is determined by finding the characteristic impedance of the source using the curves given in the input/output buffer specification (IBIS) model for the device driver which relates the inductance and capacitance of the pin and the silicon capacitance. The characteristic impedance varies slightly for each pin however all microcontroller pinouts connected to the SDRAMs were calculated to fall in the range of 35 to 45 Ω using Equation (8) as suggested in [31].

$$Z_T = \sqrt{\frac{L_{pin}}{C_{pin} + C_{comp}}} \tag{8}$$

Therefore, it was deemed best to terminate the driver's end of each transmission line so that the signals reflect off the unmatched end and terminate into the matched end as suggested in [32]. This was done by placing a series termination resistor of 22 Ω following the presented calculations at the driver's end of each trace. Data line signals being driven from both ends depending on a write or read command are terminated approximately in the middle of the line.

Other considerations were also taken in the routing and physical layout of the SDRAM chips. One-hundred nF ceramic decoupling capacitors are placed across the various power pins on the

SDRAM chips. This prevents the voltage supply from dropping when the SDRAM core requires current, as with a refresh, read or write. It also provides current during reads for the output drivers.

The number of vias on each line was also minimized in order to avoid adding extra capacitance on the traces. Also a keep out region around the microcontroller crystal was devised in order to prevent any high-speed routing across or close to the 16 MHz crystal.

### 3.2.7. Heidenhain Encoder Interface

As the end application of the instrument will be to map the magnetic field across the length of a 4-m-long undulator at PSI, the instrument supports an RS485-based interface to a Heidenhain linear absolute encoder with a resolution of 1 nm and an absolute accuracy of $\pm 3$ μm. As the instrument is mounted on a rig mechanism along the undulator length, this industrial drive requires highly reliable and low-latency position feedback. The EnDat 2.2 protocol interface from HEIDENHAIN (Traunreut, Germany) is a digital bidirectional interface standard for position or rotary encoders. The interface transmits position values and also allows reading and writing of the encoder's internal memory. The type of data transmitted, like absolute position, temperature, diagnostic parameters and others, is selected through mode commands that the EnDat 2.2 master sends to the encoder [33].

Communication over the EnDat 2.2 interface with the encoder is implemented using a hardware configurable logic block module on the microcontroller that is accessed via library functions as explained in [34]. This block generates the clock for the encoder and for the internal SPI module that acts as the slave receiver and synchronizes communication with the encoder. Cable propagation delay compensation functions are also implemented via library functions.

Subsequent electronics circuitry consisting of RS-485 transceivers transmits differential data and clock signals in half-duplex mode and provide an end termination characteristic impedance of 120 $\Omega$. The SN65HVD78 RS-485 transceivers [35] are chosen which can handle a maximum baud rate of 50 Mbps. This falls conveniently well above the maximum EnDat 2.2 protocol clock frequency of 16 MHz.

A very stringent requirement in this application is the synchronization of the magnetic field readings and the physical position reading across the undulator axis. The time duration between the falling edge of the ADC interrupt signal and the start of the encoder polling transmission command indicates the real time lag for the microcontroller to process the actual falling edge interrupt and enter in the programmed interrupt and set up the command in its registers to send to the encoder. This was determined experimentally as after the indicated lag of 8.4 μs in Figure 6 the data command is clocked out at which point the encoder saves its current position and later is clocked out and sent to the microcontroller. As the microcontroller operates at a clock frequency of 200 MHz, the microcontroller takes 1680 clock cycles to handle fully this request.

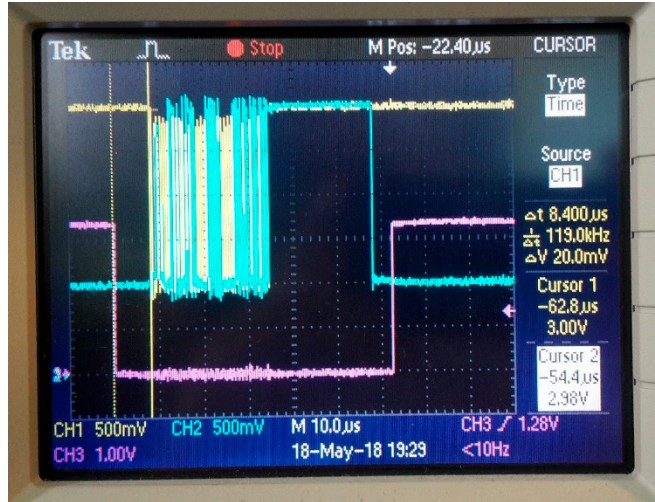

**Figure 6.** Oscilloscope snapshot showing the time delay of 8.4 µs between the falling edge of the ADC interrupt signal and the start of the encoder polling transmission command where the yellow trace is the encoder clock, blue trace is the encoder data and red trace is the ADC interrupt signal.

### 3.2.8. Voltage Regulation Circuitry

The instrument must be supplied with a ±6 V DC power supply. The 12 V across the supply rails directly feed a cooling fan placed on top of the microcontroller for convection heat extraction. The +6 V is regulated down to +5 V using the TPS7A4700 low dropout regulator [36]. This voltage rail powers the current source circuitry due to the headroom required by the op amp. The +5 V is then regulated to +3.3 V, which powers the positive rail of the remaining analogue circuitry. The +3.3 V is further regulated down to +2.5 V, which powers the positive supply rail of the analogue dynamic range of the ADC. This cascaded design of the LDOs (low dropout's) network was the preferred choice in improving drastically the power supply rejection ratio (PSRR) as enough voltage headroom is present between subsequent stages. Therefore as identical LDOs are used with a PSRR of 78 dB, the PSRR of the +2.5 V analogue supply voltage of the ADC triples to 234 dB, which provides excellent suppression of any noise and ripples from corrupting the ADC output.

The negative rail LDOs are also similarly cascaded providing comparable power supply noise performance for the analogue circuitry operating at −3.3 V and the negative ADC rail voltage of −2.5 V.

Because the total RMS supply current draw of the instrument is 300 mA and each LDO is capable of sourcing a 1 A load, negligible degradation in the PSRR is observed. The use of switching regulators for the generation of negative supply from positive supply is avoided in order to keep the system noise as low as possible.

The noise performance of the LDOs depends mostly on a noise free ground connection. Therefore, the thermal pad of each LDO is soldered directly to a pad on the PCB containing a 5 × 5 pattern of 0.25 mm vias for conducting heat. This thermal pad is also directly connected to the two internal ground planes.

### 3.2.9. Communication Interfaces: USB 2.0 and TTL Signals

The most convenient method of communication to the instrument is via a mini USB 2.0 port. The USB controller of the microcontroller operates as a full-speed function controller during point-to-point communications with the USB host. Due to the nature of data to be transferred, the bulk data transfer mode is implemented where communication is done using data bulk transfers of 512 bytes reaching a maximum transfer speed of 7 Mbit/s. Constrains in physical buffer sizes of the USB controller limit the maximum speed reached.

A bulk device class USB driver has been developed using the NI-VISA driver software by National Instruments Corporation (Austin, TX, USA) with the correct vendor ID and product ID of the instrument for device USB recognition. As the USB module is powered using a 60 MHz output clock set up by the auxiliary phase locked loop (PLL) and due to the differential signaling nature of the USB data lines, the D+ and D− traces from the microcontroller to the USB on board connector are precisely length matched to avoid any skews in data sent or received.

An external USB isolation dongle is provided with the instrument so that any ground return currents from the host side do not flow through the ground plane of the instrument. Such currents can induce ground voltage differentials that can affect the magnitude of the sensitive voltages of the analogue side. Also, isolation eliminates any voltage spikes on the ground that can occur on the host side during USB cable ejection and insertion.

Apart from control using the USB 2.0 interface, the instrument can be operated in standalone mode whereby control is exhibited only using a single transistor-transistor logic (TTL) external signal named "START/STOP".

Upon a low to high transition of this signal, the measurement process starts and a high to low transition stops the measurement process. Calibrated data is then stored on the SD Card and upon a high to low transition on the "BUSY" signal, data transfer is complete and the SD Card can be ejected. In the event of an error, the "ERROR" signal is pulled high.

These external signals are fed through an ISO7731 triple channel digital isolator [37] that provides an insulation barrier on both the supply and ground and the I/O pin of the microcontroller.

## 4. Embedded Microcontroller Software Program

The operation of the instrument is controlled by the 32-bit microcontroller. The program was developed and is stored on the flash read only memory (ROM). Upon power up, the CPU does its internal initialization routines and branches to the memory location of the starting point of the main code as defined in the linker file.

The flowchart shown in Figure 7 shows the sequential program flow that the instrument follows both in operation mode and calibration mode. The instrument defaults to operation mode after the microcontroller boots up and initializes all the on-board peripherals. Calibration mode must be chosen specifically by the user by sending a predefined instruction from the calibration LabVIEW (National Instruments Corporation, Austin, TX, USA) interface from the host PC through the USB connection. In this mode, real-time data is sent by the instrument through the USB so that the user performing calibration can graphically see real time plots and stores the data for analysis purposes.

The encoder is initialized only for operation mode and if no USB event connection is detected, the instrument enables the interrupts from the external TTL signals, otherwise these are disabled to avoid conflicting commands. In standalone mode the instrument temporary stores data on the SDRAM during measurement and after the "STOP" command is issued, the data is transferred automatically to the micro SD card. In USB connection mode, the instrument waits for a "START" command from the USB host and stores the acquired data on the SDRAM. After completion of the measurements, the user must select if data should be transferred to the USB host or on the micro SD card.

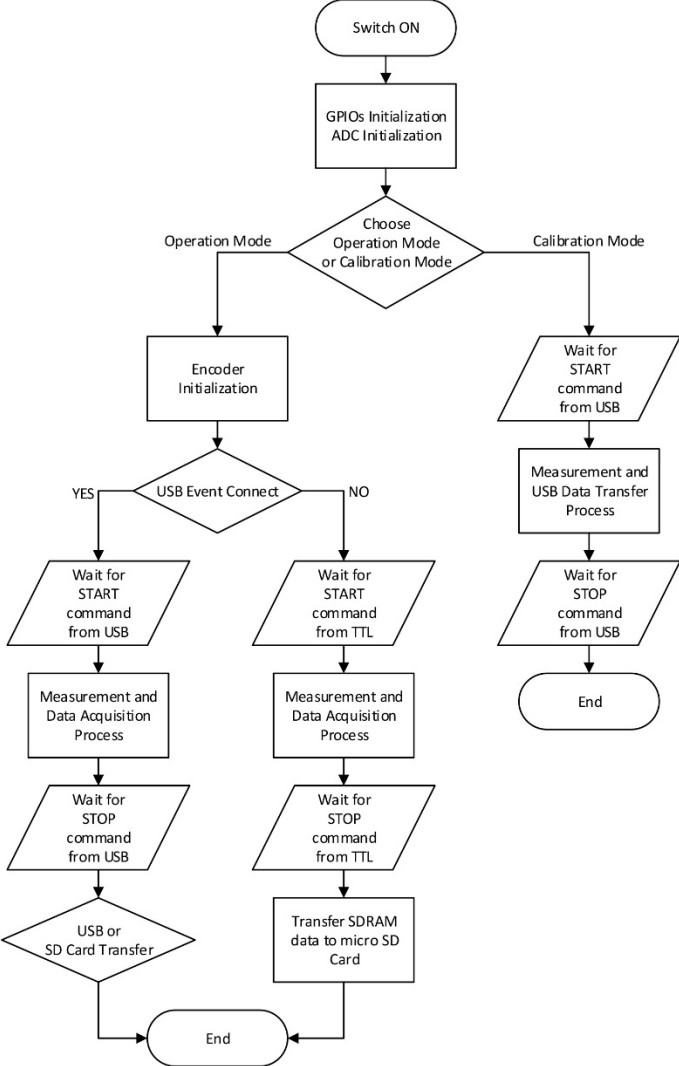

**Figure 7.** Flowchart of the instrument operation in both "Operation Mode" and "Calibration Mode".

## 5. Instrument Aluminum Enclosure

The instrument PCB is very sensitive and as both the top and bottom layers are populated with components, it must be handled with special care. A 1.6-mm-thick grey powder-coated aluminum enclosure was specifically designed and manufactured to house the PCB on 6-mm standoffs. The aluminum enclosure is also physically connected to the analogue ground plane and to the cable shielding of the Hall probe to provide electromagnetic shielding to the PCB internal to the enclosure.

A cooling fan is mounted on the inside of the enclosure to extract the heat mostly generated by the microcontroller and provide an airflow current through the enclosure for faster temperature stabilization of the electronics. Figure 8 shows the final instrument enclosure developed that houses the PCB. The air vent grille at the front serves as an air intake as the internal air circulation is extracted by the cooling fan.

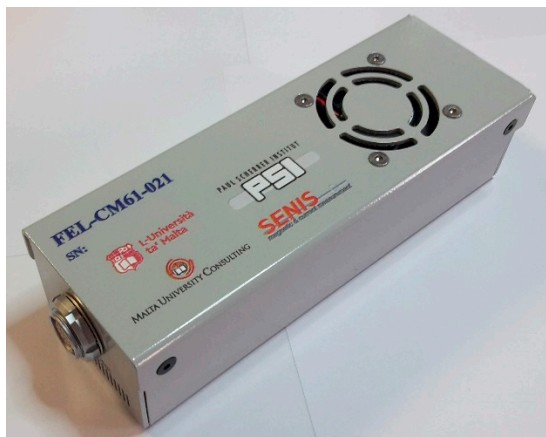

**Figure 8.** The final developed instrument with external physical dimensions of the aluminum enclosure of 150 mm by 50 mm by 45 mm.

## 6. Experimental Measurements Results

### 6.1. Magnetic Field Range Testing

The instrument testing for the ±2 T range is performed using the 7404 Lakeshore VSM electromagnet (Lake Shore Cryotronics, Westerville, OH, USA). This electromagnet is primarily used to characterize the DC magnetic properties of materials as a function of magnetic field, temperature and time. Through insertion of the Hall probe between the electromagnet poles, the sensor can be exposed to the full ±2 T range and the instrument output response noted. The specifications of the 7404 Lakeshore VSM indicate a field accuracy of ±0.05% at full scale. The full scale at an air gap of 16.2 mm is specified to be ±2.17 T resulting in an accuracy of ±1mT.

Initial experimentation is performed in order to set the correct gain of the instrument analogue amplification stage so that full dynamic range of the ADC covers ±2 T. Using standard value and high precision resistors with very low temperature coefficient, the gain of the amplification stage is set so that the ±2 T reaches 93% of maximum signal swing to avoid saturation of the ADC analogue inputs. This is ensured by setting the magnetic field strength of the electromagnet to ±2 T and tweaking the amplification voltage gain accordingly until a voltage of ±3.809 V is read. This covers 93% of the full dynamic range of the ADC (±4.096 V) and results in a final overall gain of 4.33 which gives a transduction ratio of 1.90464. Based on this ratio, noise performance results presented in the following sections are represented in Tesla rather than Volts.

### 6.2. Noise Performance

The noise performance of the whole instrument-sensor setup is determined using two figures of merits in order to quantify both the AC and the DC noise. Noise performance is characterized by the $1/f$ noise at quasi-DC measurement conditions and higher frequency noise beyond the $1/f$ corner frequency where the instrument is subject to white noise across the frequency spectrum up to the 500 Hz bandwidth of the Hall probe.

The DC resolution is given by the specification "Offset fluctuation and drift" whereas the AC resolution is given by the specification "Broadband" noise. The RMS noise voltage of the transducer in the frequency band from $f_L$ to $f_H$ is estimated by Equation (9) when combining both AC and DC noise.

$$V_{rmsB} \approx \left[ NSD_{1/f}^2 \times 1Hz \times ln\frac{f_H}{f_L} + 1.22 \times NSD_W^2 \times f_H \right]^{\frac{1}{2}} \qquad (9)$$

$NSD_{1/f}$ is the $1/f$ noise voltage spectral density at $f$ equals to 1 Hz. $NSD_W$ is the RMS white noise voltage spectral density. The numerical factor 1.22 is determined by a second-order low pass

filter. To quantify both the AC and DC noise of the instrument for the different ADC output data rate frequencies, data acquisition of readily calibrated data is done over a time period of 10 s during which time the Hall Probe is placed in a high permeability three layer Mu-metal chamber that provides a near theoretical zero gauss test volume.

### 6.3. Offset Fluctuation and Drift (0.1–10 Hz)

The DC resolution of the instrument is limited by the offset fluctuation and drift in the frequency bandwidth from 0.1 to 10 Hz. Data acquired over a 10 s period is passed through an external digital second order low-pass Butterworth filter with a bandwidth of 10 Hz so that the out-of-band noise is largely attenuated. A 10 s period is taken in order to capture the full bandwidth from 0.1 to 10 Hz. Only the first term for $NSD_{1/f}$ is computed in Equation (9) in order to find the $1/f$ noise spectral density.

The standard deviation of the offset fluctuations corresponds to the integral noise of the device in the frequency range 0.1 to 10 Hz as explained in [38] and [39]. The optimal offset fluctuation and drift response is given at an output data rate of 1 kHz with a standard deviation noise figure of 0.78 µT. This degrades to 1.26 µT at 8 kHz output data rate. These results are shown in Table 3.

**Table 3.** Standard deviation and peak to peak noise performance figures for Offset Fluctuation and Drift.

| Output Data Rate/kHz | Before 10 Hz External LPF | | After 10 Hz External LPF | | |
|---|---|---|---|---|---|
| | 1σ Error/µT | Pk-Pk error/µT | 1σ Error/µT | Pk-Pk error/µT | $NSD_{1/f}/\mathrm{µT}/\sqrt{\mathrm{Hz}}$ |
| 1 | 1.770117 | 10.620700 | 0.782971 | 4.697829 | 0.364864 |
| 2 | 2.356910 | 14.141465 | 1.094589 | 6.567535 | 0.510077 |
| 4 | 2.635093 | 15.810560 | 1.160819 | 6.964918 | 0.540940 |
| 8 | 3.133331 | 18.799990 | 1.266418 | 7.598508 | 0.590149 |

### 6.4. Broadband Noise (10 Hz–$f_T$)

The AC resolution of the instrument is given by the specification "Broadband Noise". The calibrated acquisition data for the 10 s period is passed through a second order digital band-pass Butterworth filter with low and high cut off frequencies at 10 and 500 Hz respectively. This is done in order to capture the full bandwidth of the Hall probe, which is 500 Hz and filter out the "Offset fluctuation and drift" noise component. However, the internal sinc$^3$ filter of the ADC poses a frequency upper bandwidth limitation of 262 Hz when operated at 1 kHz output data rate. For the other data rates settings the bandwidth of the sinc$^3$ filter is beyond the 500 Hz bandwidth of interest.

The white noise spectral density is computed using the second term in Equation (9). An $NSD_W$ of 0.063 µT/$\sqrt{\mathrm{Hz}}$ is achieved at an output data rate of 1 kHz which gives a standard deviation value of 1.56 µT. This defines the best AC noise performance of the instrument at a bandwidth of 262 Hz which degrades to 2.05 µT for the full sensor bandwidth of 500 Hz when operated at an output data rate of 2 kHz. Results are shown in Table 4 which are extracted from the histogram plots in Figure 9.

**Table 4.** Broadband noise performance figures for the indicated bandwidths.

| Output Data Rate/kHz | Bandwidth/Hz | 1σ Error/µT | $NSD_W/\mathrm{µT}/\sqrt{\mathrm{Hz}}$ |
|---|---|---|---|
| 1 | 262 | 1.567720 | 0.0634 |
| 2 | 500 | 2.056552 | 0.0832 |
| 4 | 500 | 2.395707 | 0.0969 |
| 8 | 500 | 2.864964 | 0.1159 |

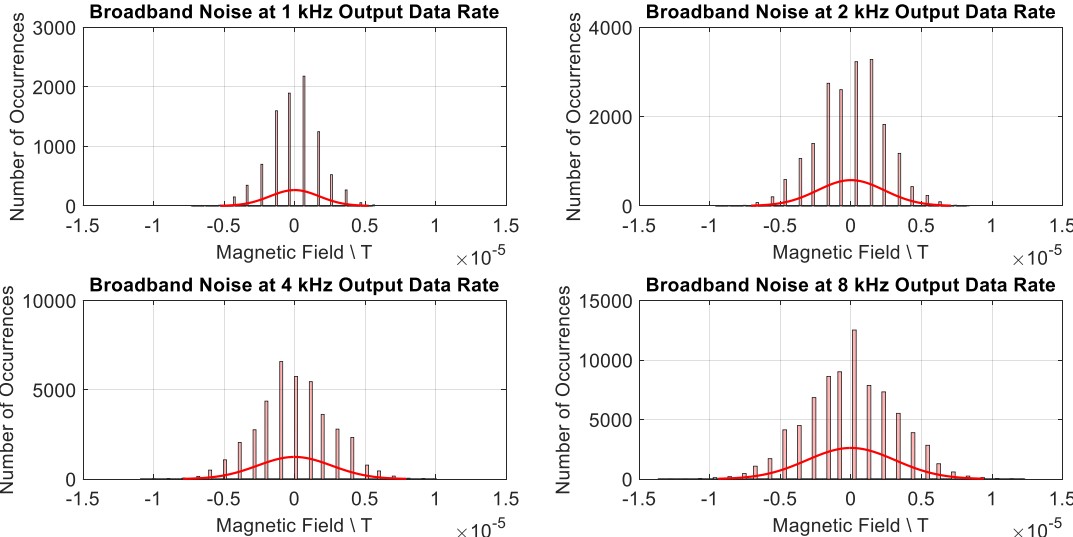

**Figure 9.** Broadband noise performance plots for the different output data rate settings. One can note that the DC component is filtered out by the external digital band pass filter.

## 7. Discussion

The theoretical magnetic resolution of the ADC of the newly developed instrument is 24 bits over the $\pm 2$ T calibration range. However, as the effective resolution of the ADC at the minimal output data rate of 1 kHz is 22.19 bits, this gives a magnetic resolution of 0.8 µT. The additional noise which results in a standard deviation of 1.2 µT is induced by the analogue interfacing circuitry, the Hall probe response and the induced noise in the Hall probe cable.

The SENIS H3A magnetic field transducer datasheet [9] has a specifications table that can be directly compared to the presented results in Section 6 for this instrument. The H3A transducer is specified to have a noise spectral density of 0.2 µT$/\sqrt{\text{Hz}}$ at $f$ = 1 Hz and a noise spectral density of 0.05 µT$/\sqrt{\text{Hz}}$ at $f$ > 10 Hz. These compare very well to the noise performance figures of the developed instrument which are 0.36 µT$/\sqrt{\text{Hz}}$ at $f$ = 1 Hz and 0.063 µT$/\sqrt{\text{Hz}}$ at $f$ > 10 Hz. It is to be noted and appreciated however, that since the SENIS H3A magnetic field transducer is fully analogue and external digitization to the instrument must be applied by an external analogue to digital data acquisition system, these noise figures only quantify the analogue noise read out as the noise spectral density measurements are done over a minimal range of $\pm 100$ mV with no amplification.

This is in contrast to the new developed instrumentation whose noise figures can be determined directly from the digital readings obtained from the on-board 24-bit ADC amplified to the $\pm 2$ T range as explained in Section 6. Hence whilst the former system only presents noise performance from the analogue stage, the latter system has comparable noise performance but includes both the analogue and the digital stage.

Bandwidth limitations are predominantly determined from the Hall probe sensor used, where highly nonhomogenous magnetic fields require integrated three-axis Hall probes whose frequency bandwidth ranges up to 75 kHz as pointed out in [39]. The SENIS Hall probe [7] used for mapping the Athos PSI line of undulators has a limited bandwidth of 500 Hz, which for this application is deemed to be sufficient. As the H3A magnetic field transducer [9] also interfaces to the same S type Hall probe, its bandwidth is also limited to 500 Hz.

Low-cost teslameters, such as the one presented in [40,41], provide a more cost-effective solution with limited range and accuracy. For example, a range of $\pm 55$ mT with an accuracy of 0.2% is covered by this teslameter. Low-cost commercially available teslameters such as this one are handheld instruments with performance relying on the limited 10-bit resolution of the built-in ADC of the MCU in this case. This makes such instrumentation unsuitable for very high precision field mapping.

Compared to these commercially available teslameters, the developed instrumentation provides a fully integrated solution with an optimized analogue-to-digital conversion stage and a proper spinning current analogue readout circuit for the interfacing of a three-axes Hall probe sensor. Hence, much higher performing circuitry is condensed in a smaller form factor volume.

Table 5 gives a summary of the main specifications of the best teslameters from three different leading vendors currently found on the market as indicated in [39]. The last column outlines the specifications of the developed teslameter presented here. It is to be noted that due to the on-chip integration of the signal conditioning electronics for the SENIS transducer, superior performance is achieved most notably in the bandwidth response. The published DC field accuracy defines the maximum difference between the actual measured magnetic flux density and that given by the teslameter. This is determined after a full calibration for the instrument is performed. The indicated DC field accuracy of 0.01% for the developed instrument is presented based on initial preliminary results after the application of nonlinearity calibration only which is modelled using a fifth order polynomial. This is deemed to be improved further by experimentation of higher order polynomials in this regard and the application of temperature calibration.

**Table 5.** General specifications summary of state of the art teslameters currently found on the market in comparison to the developed instrument. Information adapted from [39].

| Parameter | SENIS | Projekt Elektronik | Group3 | Developed Teslameter |
|---|---|---|---|---|
| DC Field Accuracy/% | 0.002% | 0.005% | 0.01% | 0.01% |
| Magnetic Resolution/µT | 0.1 µT | 0.1 µT | 0.1 µT | 0.8 µT |
| Measurement Range/T | 1 µT–30 T | 20 mT–2T | 0.3–3T | ± 2 T |
| Bandwidth/kHz | DC–75 kHz | DC–1 kHz | DC–3 kHz | DC–0.5 kHz |

An additional novelty of the developed electronic module is the quasi simultaneous measurement of both the magnetic field in the three axes and the absolute linear encoder position with only an 8.4 µs delay lag. Even though the instrument has been developed as a general instrument that can be used for other machines, its specification has been developed, first and foremost, for the ATHOS soft X-Ray beamline at PSI in order to achieve a true magnetic field map across the length of its undulators. When considering that the developed instrument has similar performance to the state-of-the-art teslameters [39] but with a much smaller form factor, an integrated digitization stage, integrated position reading capability and ultimately a cheaper solution—one can appreciate that this new instrument offers several advantages over the state of the art for use in other machine undulator mapping applications and even general purpose ones.

Additional research is currently being conducted in order to devise the best calibration methods for nonlinearity calibration, temperature changes compensation and Hall probe angular errors correction.

Temperature compensation will surely include offset and sensitivity compensation over the full ±2 T dynamic range for the precise modelling of the effect of the temperature changes on the magnetic field readout. As indicated in [2], the offset and sensitivity of the Hall device drift over temperature and this varies randomly from one device to another. These drifts are mostly caused by piezo-resistive effects. Also, the mechanical stress induced by thermal expansion of the material results in such effects.

The influence of temperature compensation of sensitivity will have a much greater effect than the compensation for the temperature offset only at zero Gauss. Temperature compensation of sensitivity will model the temperature effect at different magnetic field values across the whole range. Modelling this effect will include second or higher order polynomials in order not to degrade the accuracy obtained after non linearity calibration. Also, as this Hall probe is specified to have an orthogonality error of <2°, this will also be calibrated.

## 8. Conclusions

The developed high-accuracy teslameter can measure magnetic fields in the range of ±2 T. The instrument was designed within the required specifications set by the PSI undulator characterization

and provides digitized readings of the three magnetic field axes, the PT100 and the absolute encoder position readings—all synchronized together.

Rigorous testing is performed using output data rates ranging from 1 to 8 kHz with the best noise performance results achieved at 1 kHz with a standard deviation of 1.56 μT at a limited bandwidth of 262 Hz. Measures for resolution are given by the offset fluctuation and drift with a noise spectral density of 0.36 $\mu T/\sqrt{Hz}$ and a broad band noise spectral density of 0.06 $\mu T/\sqrt{Hz}$.

Work is underway to establish an optimized and standardized calibration setup for the instrument in order to maximize its performance and measurement accuracy in micro Tesla.

**Author Contributions:** Formal analysis, J.C.; Investigation, J.C.; Methodology, J.C.; Software, J.C.; Supervision, A.S. and N.S.; Validation, A.S., N.S. and M.C.; Writing—original draft, J.C.; Writing—review & editing, A.S., N.S., M.C., S.D. and R.S.P.

**Funding:** This research received no external funding.

**Acknowledgments:** The authors would like to thank Reuben Debono for his useful guidance and help in the PCB assembly of the instruments at the Electronic Systems Lab at the Faculty of Engineering at University of Malta. The authors would like to thank R. Ganter, project leader of the Athos undulator beamline and H-H. Braun, SwissFEL machine director, for their constant support throughout the entire project. The authors would like to thank Sasa Spasic and his team at Sentronis facilities for their fruitful discussions and their guidance during testing.

**Conflicts of Interest:** The authors declare no conflicts of interest.

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
