# Peer review of "Design and Development of a Reduced Form-Factor High Accuracy Three-Axis Teslameter"

_electronics, doi:10.3390/electronics8030368_

Round 1

Reviewer 1 Report

A paper presents design and development problems of reduced form-factor high accuracy three-axis teslameter.

Authors say, that designed and developed by them instrument can be used for high fidelity characterization and optimisation of the undulators. In their opinion the instrument incorporates analogue signal conditioning for the 3 axes interface. This is something new in contrasts to older instruments which only comprise the analogue circuitry with digitization. The article presents design and development of different parts of the teslameter.

The topic of the paper is important and actual. The measurement of magnetic field density is important and necessary in many technical (and others) applications. Anyway, there is still problem with accuracy and the influence of many things on the result of the measurement. So, I think, the paper is interesting.

51 – there is some mistake in literature. First, there are 1, 2, 3 literature positions, and suddenly there is position number 34 (?). Please clarify.

I think, the introduction chapter is too short, too small. There are many things, which should be explained in this chapter. First of all – state of knowledge. Second, please explain in details what kind of problem authors want to solve. There could be more literatures positions, too. Please complete this chapter.

Authors say in introduction chapter, that measurement results depend on many factors such as temperature. I did not find any results in the paper according to the influence of temperature on a new version of describing teslameter. If there is no result, please put some comments, suggestions, how the influence can look like.

Author Response

Reviewer 1 Reply

Dear Reviewer,

Thanks for your feedback. It has been greatly appreciated. Please see my answers in green below.

Open Review

English language and style

( ) Extensive editing of English language and style required 
( ) Moderate English changes required 
(x) English language and style are fine/minor spell check required 
( ) I don't feel qualified to judge about the English language and style 

Yes

Can be improved

Must be improved

Not applicable

Does the introduction provide   sufficient background and include all relevant references?

( )

( )

(x)

( )

Is the research design appropriate?

( )

(x)

( )

( )

Are the methods adequately described?

(x)

( )

( )

( )

Are the results clearly presented?

(x)

( )

( )

( )

Are the conclusions supported by the   results?

(x)

( )

( )

( )

Comments and Suggestions for Authors

A paper presents design and development problems of reduced form-factor high accuracy three-axis teslameter.

Authors say, that designed and developed by them instrument can be used for high fidelity characterization and optimisation of the undulators. In their opinion the instrument incorporates analogue signal conditioning for the 3 axes interface. This is something new in contrasts to older instruments which only comprise the analogue circuitry with digitization. The article presents design and development of different parts of the teslameter.

The topic of the paper is important and actual. The measurement of magnetic field density is important and necessary in many technical (and others) applications. Anyway, there is still problem with accuracy and the influence of many things on the result of the measurement. So, I think, the paper is interesting.

51 – there is some mistake in literature. First, there are 1, 2, 3 literature positions, and suddenly there is position number 34 (?). Please clarify.

Yes I have modified the sequence of the presented references so that these are listed according to the way they are referenced throughout the paper.

I think, the introduction chapter is too short, too small. There are many things, which should be explained in this chapter. First of all – state of knowledge. Second, please explain in details what kind of problem authors want to solve. There could be more literatures positions, too. Please complete this chapter.

I have lengthened the introduction section by including a description of the techniques used in the measurement of magnetic field in undulators. This sets a more concrete context to the problem definition of the project whereby the developed instrument is mainly used in this scenario. References to additional papers have been included in order to give the reader more reference to present literature covering this research area.

Authors say in introduction chapter, that measurement results depend on many factors such as temperature. I did not find any results in the paper according to the influence of temperature on a new version of describing teslameter. If there is no result, please put some comments, suggestions, how the influence can look like.

As indicated in the conclusion of the paper, a standardized calibration setup and procedure is still currently being developed so no final results can yet be presented. However second and third order polynomial modelling for temperature sensitivity are being experimented with in order to model accurately the temperature influence on the Hall probe. This is explained at the end of the Discussion section.

 Thank you

Reviewer 2 Report

In my opinion it is a great article and generates a positive advance in the area of the High Accuracy Three-Axis Teslameter.
 These advances can be applied in various fields of physics and engineering.The readers of the magazine have a great reference with this article in this field.

It can be published as presented.

 I have only minor doubts in the following lines:

  1.-Line 97 ---> of of ?

  2.- Line 116--> when you say  ..."for optimal EMC performance"... How do you quantify that optimum?. What measures are taken to ensure the best value of the EMC performance ?.

  3.- Line 174--> you say ..."a tradeoff was found experimentally in determining the best excitation DC current".. How the changing ambient temperature around the sensor affects the thermal balance for this DC curren, and therefore in the temperature of the sensor in the thermal equilibrium ?

  Thank you

Author Response

Reviewer 2 Reply

Dear Reviewer,

Thanks for your feedback. It has been greatly appreciated. Please see my answers in green below.

Open Review

English language and style

( ) Extensive editing of English language and style required 
( ) Moderate English changes required 
( ) English language and style are fine/minor spell check required 
(x) I don't feel qualified to judge about the English language and style 

Yes

Can be improved

Must be improved

Not applicable

Does the introduction provide   sufficient background and include all relevant references?

(x)

( )

( )

( )

Is the research design appropriate?

(x)

( )

( )

( )

Are the methods adequately described?

(x)

( )

( )

( )

Are the results clearly presented?

(x)

( )

( )

( )

Are the conclusions supported by the   results?

(x)

( )

( )

( )

Comments and Suggestions for Authors

In my opinion it is a great article and generates a positive advance in the area of the High Accuracy Three-Axis Teslameter. 
 These advances can be applied in various fields of physics and engineering.The readers of the magazine have a great reference with this article in this field.

It can be published as presented.

 I have only minor doubts in the following lines:

1.-Line 97 ---> of of ?

Yes this typo has been corrected.

  2.- Line 116--> when you say  ..."for optimal EMC performance"... How do you quantify that optimum?. What measures are taken to ensure the best value of the EMC performance ?.

I have slightly altered this paragraph in order to make it more clear what I meant. The choice of the word “optimum” was incorrectly chosen as the whole point here is that the choice of the 8 layer stack up enabled us to use two ground planes which are shielding the two internal signal layers. This is in comparison to a 6 layer stack up with a single ground plane. Thus the 8 layer stack up provides superior EMC performance in comparison to a 6 layer stack up. This was not actually quantified as a 6 layer version of the instrument was never manufactured

3.- Line 174--> you say ..."a tradeoff was found experimentally in determining the best excitation DC current".. How the changing ambient temperature around the sensor affects the thermal balance for this DC current, and therefore in the temperature of the sensor in the thermal equilibrium?

The thermal balance of the bias current through the Hall probe is dependent on the thermal coefficient of the driving biasing current source circuitry. This Howland Current source based design relies on a 2.5V reference with a temperature coefficient of 3 ppm/°C and a 1 kΩ resistor array with a temperature coefficient of 10 ppm/°C. Therefore predominantly the ambient temperature of the electronics affects the DC current magnitude through the Hall probe with the mentioned temperature coefficients rather than the temperature of the Hall probe itself. The temperature of the Hall probe itself has a direct effect on the probe’s resistance which needs to be compensated for through proper calibration as indicated in the last part of the “Discussion” section in the paper.

Thank you

Thank you

Reviewer 3 Report

Thank you for your submission. The manuscript is interesting and worthy of publication. It is very well structured, and clear. The submission is an excellent electronic system engineering manuscript. As such, I believe it is appropriate to be published.

As a downside, I think that the manuscript does not present a high level of novelty, in that it presents an improvement of an existing instrument.

I would ask to improve the paper by adding a detailed comparison with state-of-the-art instruments, to see how the presented device is positioned in terms of the relevant parameters.

Author Response

Reviewer 3 Reply

Dear Reviewer,

Thanks for your feedback. It has been greatly appreciated. Please see my answers in green below.

Open Review

English language and style

( ) Extensive editing of English language and style required 
( ) Moderate English changes required 
(x) English language and style are fine/minor spell check required 
( ) I don't feel qualified to judge about the English language and style 

Yes

Can be improved

Must be improved

Not applicable

Does the introduction provide   sufficient background and include all relevant references?

( )

(x)

( )

( )

Is the research design appropriate?

(x)

( )

( )

( )

Are the methods adequately described?

(x)

( )

( )

( )

Are the results clearly presented?

(x)

( )

( )

( )

Are the conclusions supported by the   results?

(x)

( )

( )

( )

Comments and Suggestions for Authors

Thank you for your submission. The manuscript is interesting and worthy of publication. It is very well structured, and clear. The submission is an excellent electronic system engineering manuscript. As such, I believe it is appropriate to be published.

As a downside, I think that the manuscript does not present a high level of novelty, in that it presents an improvement of an existing instrument.

Yes the developed instrument is an improvement over the old instrumentation setup as described in the abstract of the paper. This improvement results in a much more compact instrument that includes both the analogue and digital circuitry, on board software calibration and a synchronized interface to an encoder. As for the old instrument, off the shelf products were used. For this reason, a whole redesign of the setup had to be done, resulting in the novelty of the tailored solution presented in this paper.

I would ask to improve the paper by adding a detailed comparison with state-of-the-art instruments, to see how the presented device is positioned in terms of the relevant parameters.

A comparison with state of the art teslameters is presented in the “Discussion” section where additional detail has been included in the presented table that outlines a direct comparison of the parameters of the developed instrument to three major providers of teslameters.

Thank you

Round 2

Reviewer 1 Report

1. I said that there is some mistake in literature in first version of the paper. Authors have modified the sequence of the presented references.

2. I concluded that introduction chapter in first version of the paper is too short, too small. There are many things, which should be explained in this chapter. First of all – state of knowledge. Authors have lengthened the introduction section by including a description of the techniques used in the measurement of magnetic field in undulators

3. I wrote that authors say in introduction chapter of first form of the paper, that measurement results depend on many factors such as temperature. I did not find any results in the first version of the paper according to the influence of temperature on a new version of describing teslameter. Authors answer that second and third order polynomial modelling for temperature sensitivity are being experimented with in order to model accurately the temperature influence on the Hall probe.

I think, all my comments are included in the text. In my opinion, the paper is ready to be published in present form.